# CONSERVATIVE REINFORCEMENT LEARNING BY Q-FUNCTION DISAGREEMENT

## ABSTRACT

In this paper we propose a novel continuous-space RL algorithm that subtracts the Q-target network standard deviation from a Q-target network which leads to forcing a tighter upper-bound on Q-values estimation. We show in experiments that this novel Q-target formula has a performance advantage when applied to algorithms in this space such as TD3, TD7, MaxMin, REDQ, etc., where the domains examined are control tasks from MuJoCo simulation. We provide the code in `https://github.com/anonymouszxcv16/SQT`.

## 1 INTRODUCTION

Reinforcement learning (Sutton & Barto, 2018a; Bertsekas, 2005; Mannor et al.) has gained a lot of traction in solving control problems where the model is missing and the action space is continuous. In recent years, several algorithms were proposed to solve these problems, such as TD3 (Fujimoto et al., 2018), TD7 (Fujimoto et al., 2023), CQL (Kumar et al., 2020), REDQ (Chen et al., 2021), MaxMin Q-learning (Lan et al., 2020), REM (Agarwal et al., 2020), and more.

A technique used in these algorithms is using the so-called "Double Q-learning" (Hasselt, 2010) which stabilizes the algorithm. In addition, it is proved in this technique that both its Q-functions converge to a single fixed point. In that case, the mean of both networks Q-functions standard deviations must be 0 when the algorithm converges. In this work, we use this insight to propose an algorithm that depends on this phenomenon in order to incorporate a conservative estimation of the Q-value.

From a different point of view, TD3 states clearly that the Double Q-learning Q-target update-rule introduces some bias into the Q-functions' estimations. The TD3 algorithm tackles this problem by taking the minimum between the two Q-functions as a wide-consensus estimation.

Our main contributions in this paper are as follows:

1. We provide a novel regularization scheme for updating the Q-target formula, based on the two networks maintained by the Double Q-learning technique.
2. We show in experiments how this technique leads to faster convergence when added to recent SOTA algorithms.

The paper is structured as follows. We begin with describing the setup in Section 2. We review related work in Section 3. Then, we state our main algorithm in Section 4 and in Section 5 we conduct experiments to demonstrate the advantage of our approach. We conclude in Section 6 with discussion, conclusions, and future work.

## 2 SETUP

We model the problem using a Markov Decision Process (MDP; Puterman (1994)), where $\mathcal{S}$ and $\mathcal{A}$ are the state space and action space, respectively. We let $P(s'|s,a)$ denote the probability of transitioning from state $s \in \mathcal{S}$ to state $s' \in \mathcal{S}$ when applying action $a \in \mathcal{A}$. We consider a probabilistic policy $\pi_\phi(a|s)$, parameterized by $\phi \in \Phi$ which expresses the probability of the agent to choose an action $a$ given that it is in state $s$. The reward function is denoted by $r(s,a)$. The

goal of the agent is to find a policy that maximizes the *Q-function* that the agent receives during its interaction with the environment Puterman (1994).

$$Q_\phi(s,a) \triangleq \mathbb{E}\left[\sum_{t=0}^{\infty} r(s_t, a_t)|s_0 = s, a_0 = a\right]. \tag{1}$$

## 3 RELATED WORK

In this section we review related work upon which we base our approach.

**Actor Critic Methods.** Actor Critic methods Konda & Tsitsiklis (1999); Sutton & Barto (2018b); Castro & Meir (2010) have two components. One component is an *actor* that learns a policy $\pi : \mathcal{S} \to \mathcal{A}$, mapping state to action. The second component is a *critic* that learns a state-action pair estimation $Q(\cdot, \cdot) : \mathcal{S} \times \mathcal{A} \to \mathbb{R}$.

**Double Q-learning.** In stochastic environments Q-learning Watkins & Dayan (1992) may perform poorly, mainly in terms of stability due to its bootstrap nature. This poor performance is caused by over-estimations of action values resulting from a positive bias that is introduced because Q-learning uses the maximum action value as an approximation for the maximum expected action value. Hasselt (2010) proposed *Double Q-learning*. This method stores two Q-functions denoted by $Q^A(\cdot, \cdot)$ and $Q^B(\cdot, \cdot)$. Each of these Q-functions is updated with a value from the other Q-function for the next state. Let us denote the action $a^*$ to be the maximal valued action in state $s'$ according to $Q^A$. Next, instead of using the value $Q^A(s', a^*) = \max_a Q^A(s', a)$ to update $Q^A$, as Q-learning does, Double Q-learning uses the value $Q^B(s', a^*)$. Since $Q^B$ was updated on the same problem with different experience samples – it is unbiased w.r.t. action $a^*$. Interchanging the roles of the two networks yields the following updates:

$$Q^A(s,a) \leftarrow Q^A(s,a) + \alpha(s,a)(r + \gamma Q^B(s', a^*) - Q^A(s,a)), \quad a^* = \arg\max_a Q^A(s', a)$$
$$Q^B(s,a) \leftarrow Q^B(s,a) + \alpha(s,a)(r + \gamma Q^A(s', b^*) - Q^B(s,a)), \quad b^* = \arg\max_a Q^B(s', a). \tag{2}$$

**Deep Deterministic Policy Gradient algorithms.** Deterministic Policy Gradient (DPG; Silver et al., 2014) algorithm uses an expected gradient of the action-value function $Q : \mathcal{S} \times \mathcal{A} \to \mathbb{R}$. This gradient can be estimated much more efficiently than the usual stochastic policy gradient Sutton & Barto (2018b). DPG introduces an off-policy actor-critic algorithm that learns a deterministic target policy $\pi$ from an exploratory behavior policy denoted by $\mu$. Deep Deterministic Policy Gradient (DDPG; Lillicrap et al. (2015)) incorporates a deep learning approach into DPG. Since it is not possible to straightforwardly apply Q-learning Watkins & Dayan (1992) to continuous action spaces, finding the greedy policy requires optimization at every time step. This is too slow to be practical with large function approximators. Instead, DDPG uses an actor-critic architecture based on DPG where the gradient is estimated based on the following formula

$$\nabla_{\theta^\mu} J \approx \mathbb{E}_{s_t \sim \rho^\beta}[\nabla_a Q(s, a|\theta^Q)|_{s=s_t, a=\mu(s_t)} \nabla_{\theta_\mu} \mu(s|\theta^\mu)|_{s=s_t}]. \tag{3}$$

**Twin Delayed DDPG (TD3).** The TD3 algorithm (Fujimoto et al., 2018) is an algorithm that is based on DDPG with several additional tricks. First, it has a clipped action (or *target policy smoothing*), i.e., the action applied is

$$a(s) = \text{clip}\left(\mu(s) + \text{clip}\left(\epsilon, -c, c\right), a_{\text{low}}, a_{\text{high}}\right), \quad \epsilon \sim \mathcal{N}(0, \sigma). \tag{4}$$

Essentially, this is a regularizer for the action taken, mitigating the risk of taking action that is too sharp. Second, it has a mechanism to reduce overestimation of the Q-value. This is done with considering the more conservative estimate of the Q-value:

$$y_1 = r + \gamma \min_{i=1,2} Q_{\theta'_i}(s', \pi_{\phi_1}(s')). \tag{5}$$

We note that the optimization on the Q-function is done similarly to DDPG.

**TD7 Algorithm**. The TD7 algorithm (Fujimoto et al., 2023) is a TD3 algorithm with 4 additions.

1. **State Action Learned Embeddings (SALE)** Fujimoto et al. (2023) learns (similarly to Ota et al., 2020) an embedding for the state-action and state $(z^{sa}, z^s)$ that tries to capture the transition representations, i.e.,

$$z^{sa} := g(z^s, a), \quad z^s := f(s),$$

where $g(\cdot, \cdot)$ and $f(\cdot)$ are the encoders satisfying the minimization of the following loss

$$\mathcal{L}(f, g) = (g(f(s), a) - f(s')),$$

where $s'$ denotes the next state. Practically, TD7 concatenates the embedding with the state-action for the Q-function evaluation:

$$Q(s, a) \rightarrow Q(z^{sa}, z^s, s, a).$$

In addition, TD7 concatenates the embedding with the state for the policy evaluation, i.e.,

$$\pi(s) \rightarrow \pi(z^s, s). \tag{6}$$

Therefore, the TD7 target formula becomes

$$y = r + \gamma \min_{i=1,2} Q_{\theta'_i}(s', a', z^{s'a'}_{\rho''}, z^{s'}_{\psi''}), \quad a' \leftarrow \pi_{\phi'}(s', z^{s'}_{\psi''}) + \epsilon, \quad \epsilon \sim \mathcal{N}(0, 1), \tag{7}$$

and the actor update-rule is

$$\nabla_\phi J \approx \frac{1}{|\mathcal{B}|} \sum_{(s,a) \in \mathcal{B}} \frac{1}{2} \sum_{i=1,2} [Q_{\theta_i}(s, a, z^{sa}_{\rho'}, z^s_{\psi'})], \quad a = \pi_\phi(s, z^s_{\psi'}). \tag{8}$$

2. **Prioritized Experience Replay Buffer (PER)**. Unlike vanilla replay buffer where all samples are uniformly sampled, TD7 uses the sampling method presented by Schaul et al. (2015) and Fujimoto et al. (2020) (Loss-Adjusted Prioritized replay buffer) which samples batch according to the TD error. Formally,

$$p(i) = \frac{\max(|\delta(i)|^\alpha, 1)}{\sum_{j \in \mathcal{D}} \max(|\delta(j)|^\alpha, 1)}, \quad \delta(i) := Q(s, a) - (r + \gamma Q(s', a')).$$

3. **Clip**: clips Q-target by the minimal and maximal Q-values:

$$Q_{t+1}(s, a) \approx r + \gamma \text{clip}(Q_t(s', a'), \min_{(s,a) \in \mathcal{D}} Q_t(s, a), \max_{(s,a) \in \mathcal{D}} Q_t(s, a))$$

4. **Checkpoints**. Checkpoints are a technique to use the best-performing policy for evaluation. Deep RL algorithms are notoriously unstable Fujimoto et al. (2023). In supervised learning checkpoints are a common approach for selecting high-performing models and maintaining a consistent performance across evaluations Devlin et al. (2019). A checkpoint is a snapshot of the parameters of a model captured at a specific time during training. In RL using the checkpoint of a policy that obtained a high reward during training instead of the current policy improves the stability of the performance at test time.

**Randomized Ensembled Double Q-Learning Algorithm** (REDQ; Chen et al., 2021) is an algorithm which maintains $N$ networks (typically more than 2), samples a subset of $\mathcal{M}$ Q-functions, and then takes the minimal Q-function from this subset reducing weighted $\log$ policy as its Q-target formula. Specifically,

$$y = r + \gamma(\min_{i \in \mathcal{M}} Q_{\phi_{targ,i}}(s', \tilde{a}') - \alpha \log \pi_\theta(\tilde{a}'|s')), \quad \tilde{a}' \sim \pi_\theta(\cdot|s'), \quad \mathcal{M} \sim \{1, ..., N\}.$$

We note that in a sense, this algorithm is a generalization of target from 2 networks to $N$ networks.

**MaxMin Q-learning** Lan et al. (2020) proposed to take the minimal Q-function between the $N$ functions as its Q-target formula

$$y = r + \gamma \cdot \min_{i \in \{1, ..., N\}} Q_{\phi_{targ,i}}(s', \tilde{a}'), \quad \tilde{a}' \sim \pi_\theta(\cdot|s'). \tag{9}$$

We note that the main difference between MaxMin Q-learning and REDQ is that one randomizes between the $N$ networks, and one takes the most conservative out of these $N$ networks. We note that by structure, these two methods are quite computationally heavy since we maintain $N$ replicates of the network.

# 4 THE STD Q-TARGET METHOD

In this section we present the *Std Q-target regularization* (SQT). This variation can be applied to any algorithm that maintains more than one target network.

We begin with defining the std of a batch. Let us consider a batch $\mathcal{B}$. We denote with $\text{std}(s', a')$ to be the empirical standard deviation of the state-action $(s', a')$ Q-value. Practically, in our case, since there are only two networks, this std is based on two samples $Q_{\theta'_i}(s', a')$ for $i = 1, 2$. Formally, we denote this std with

$$\text{std}(s', a') = \text{std}_{i=1,2}[Q_{\theta'_i}(s', a')].$$

We define the std of the batch, which is the mean of these stds, to be

$$\text{std}[\mathcal{B}] = \frac{1}{|\mathcal{B}|} \sum_{(s',a') \sim \mathcal{B}} \text{std}_{i=1,2}[Q_{\theta'_i}(s', a')]. \tag{10}$$

Based on these definitions, we introduce the Std Q-target formula

$$y = r + \gamma \min_{i=1,2} Q_{\theta'_i}(s', a') - \alpha \cdot \text{std}[\mathcal{B}], \tag{11}$$

where $\alpha$ is a meta-parameter for the regularization term. The action is chosen according to

$$a' \leftarrow \pi_{\phi'}(s') + \epsilon, \quad \epsilon \sim \mathcal{N}(0, 1). \tag{12}$$

The intuition behind equation 10 is as follows. If the Q-value estimation of the two networks is similar, the std will be relatively small. Therefore, we argue that Q-value estimation is relatively correct. If both networks disagree on the estimation, we would like to provide a conservative estimation for the Q-value, proportional to the "disagreement", i.e., the std.

**Remark 4.1.** We note that in this case we do the std estimation based on two networks, but generalization to $N$ networks is straight forward. The trade-off is between more precise std estimation and computation and space requirements.

**Remark 4.2.** Double Q-learning Hasselt (2010) proved that the two Q-functions of eq. 2 converge to a single optimal policy fixed point. We argue that the two Q-functions' standard deviation must be 0 when the iterations converge, i.e.,

$$\lim_{t \to \infty} \mathbb{E}_{s' \sim \mathcal{B}}[\text{std}_{i=1,2}[Q_{\theta'_i}(s', a')]] = 0, \quad a' = \pi_{\phi'}(s'). \tag{13}$$

Proving formally this is beyond the scope of this work but the intuition above may assist in this case: if the algorithm converges, then the std must be 0. Furthermore, to obtain a tighter upper bound of the real Q-target we can reduce the Q-functions standard deviation from the minimal Q-value, pushing the algorithm to the true value estimation.

**Remark 4.3.** It is apparent that taking the minimal Q-function (which reflects a wide consensus value) and reducing the Q-functions standard deviation (which reflects reducing the unknown estimation area for safe values) provides a conservative estimation.

**Remark 4.4.** It is easy to see that this novel Q-target formula can be successfully integrated into TD3, REDQ, MaxMin Q-learning, and REM Fujimoto et al. (2018); Chen et al. (2021); Lan et al. (2020); Agarwal et al. (2020).

A generic RL algorithm of how to incorporate the SQT regularizer is presented in Algorithm 4.

---

**Algorithm 1** Std Q-Target (SQT)

---

1: **for** each iteration $t$ **do**
2:     Observe state $s$
3:     Select action $a$.
4:     Take a step $a$ in state $s$ and receive state next $s'$ reward $r$ and terminal signal $d$
5:     Store tuple $\mathcal{D} \leftarrow \mathcal{D} \cup \{(s, a, r, s', d)\}$
6:     Sample batch $\mathcal{B} \sim \mathcal{D}$
7:     Compute batch std

$$\text{std}[\mathcal{B}] = \frac{1}{|\mathcal{B}|} \sum_{(s',a') \sim \mathcal{B}} \text{std}_{i=1,2}[Q_{\theta'_i}(s', a')]. \tag{14}$$

8:     Compute Q-target $y$ by

$$y = r + \gamma \min_{i=1,2} Q_{\theta'_i}(s', a') - \text{std}[\mathcal{B}]. \tag{15}$$

9:     Update critic parameters.
10:    Update actor parameters.
11: **end for**

---

## 5 EXPERIMENTS

We test our Std Q-target regularization on SOTA continuous-space RL algorithms: TD3 (Fujimoto et al., 2018), TD7 (Fujimoto et al., 2023), MaxMin (Lan et al., 2020), and REDQ (Chen et al., 2021). The environments we test against are MuJoCo (Todorov et al., 2012) and D4RL Fu et al. (2020) where we chose these environments since they are among the most popular simulated environments with continuous action space for the online settings (MuJoCo) and for the offline settings (D4RL; Levine et al., 2020).

For the online experiments, we tested against 7 types of environments: (1) "Humanoid" (Tassa et al., 2012) where the goal for a human like skeleton to walk as fast as it can; (2) "Walker2D" where a pair of legs need to walk forward; (3) "Ant" environment where the goal is to reach a certain speed with ant like skeleton; (4) "HalfCheetah" environment, where the task is also to reach a certain speed with a 2 leg cheetah shape; (5) "Hopper" where a single leg needs to advance by jumping; (6) "HumanoidStandup" where a skeleton like the "Humanoid" need to stand from lying on the ground starting pose; and (7) "Swimmer" where a 3 parts snake-like needs to advance on the ground. For the offline experiments we tested against the D4RL environments ("medium" dataset) which are subset of the online experiments, i.e., it contains only the Walker2d, Ant, HalfCheetah, and the Hopper environments.

The offline environments are offline samples gathered from a sub-optimal controller. Practically, we added to the actor loss the Behavior Cloning loss (BC; Fujimoto & Gu, 2021; Hussein et al., 2017). We note that TD3 and TD7 are already come with a BC loss while we added for MaxMin and REDQ the same loss, i.e.,

$$loss_{\text{BC}} = \sum_{(s,a') \in \mathcal{B}} [a - a']^2, \quad a = \pi_\phi(s).$$

We note that we measure the success of an algorithm by the number of iterations needed for convergence. Since our contribution is a lightweight regularization added to the target network, we compared each algorithm with and without the std term of Eq. equation 11. We ran all algorithms and environments with $1e6$ iterations, and averaged over 5 seeds for each of the algorithm-environment settings. We note that due to some initial stability issues, we started the experiments in several cases with $\alpha = 0$ where at some point we increases it to a value between 0 to 1. We enlist in the Appendix the optimal values for that.

The results are presented in Tables 1-4. On average, the performance over both online and offline environments increased in $0.9\%$. It is apparent that in some cases we have significant improvement like in the TD7 and TD3 cases, while for MaxMin and REDQ there is a slight degradation.

Averaging over all the online settings, we observe an increase of $3.14\%$ in performance while for the offline settings we observe a decrease of $2.43\%$ in performance. We show the dynamics of the comparisons in Figures (1)-(7).

We argue that in TD7 our estimations for the Q-functions are more accurate. Therefore, if there is a disagreement between the networks the regularization is most informative. In TD3, which can be considered as a limited TD7, the estimation is noisier. Therefore, the regularization is less effective. Observing the results for MaxMin and REDQ, we notice that for ensemble approaches, the impact of the std regularization is limited.

| Environment | TD3 | TD3+SQT | Improv. |
|---|---|---|---|
| Humanoid (MuJoCo) | $4{,}715.3 \pm 514.1$ | $\mathbf{6{,}423.7} \pm 137.9$ | **+36%** |
| Walker2d (MuJoCo) | $5{,}421.5 \pm 93.6$ | $\mathbf{5{,}446.0} \pm 141.8$ | **+0.6%** |
| Ant (MuJoCo) | $\mathbf{6{,}353.9} \pm 64.4$ | $5{,}703.4 \pm 158.6$ | -11% |
| HalfCheetah (MuJoCo) | $12{,}086.7 \pm 188.3$ | $\mathbf{12{,}639.8} \pm 85.5$ | **+4%** |
| Hopper (MuJoCo) | $\mathbf{3{,}376.5} \pm 74.7$ | $2{,}885.4 \pm 255.6$ | -15% |
| HumanoidStandup (MuJoCo) | $161{,}214.0 \pm 6{,}925.7$ | $\mathbf{162{,}898.2} \pm 965.0$ | **+1%** |
| Swimmer (MuJoCo) | $\mathbf{134.1} \pm 2.3$ | $126.0 \pm 3.5$ | -6% |
| Walker2d (D4RL) | $53.6 \pm 9.8$ | $\mathbf{56.6} \pm 5.2$ | **+5%** |
| Ant (D4RL) | $119.7 \pm 3.3$ | $\mathbf{120.3} \pm 1.8$ | **+0.5%** |
| HalfCheetah (D4RL) | $55.4 \pm 1.2$ | $\mathbf{57.7} \pm 0.1$ | **+4%** |
| Hopper (D4RL) | $\mathbf{75.1} \pm 2.6$ | $70.3 \pm 1.1$ | **-6.4%** |
| | | | **+1.15%** |

Table 1: Comparison between TD3 and TD3+SQT.

| Environment | TD7 | TD7+SQT | Improv. |
|---|---|---|---|
| Humanoid (MuJoCo) | $6{,}653.3 \pm 110.3$ | $\mathbf{7{,}016.0} \pm 328.9$ | **+5%** |
| Walker2d (MuJoCo) | $6{,}057.8 \pm 164.3$ | $\mathbf{6{,}974.5} \pm 116.2$ | **+15%** |
| Ant (MuJoCo) | $7{,}369.3 \pm 122.5$ | $\mathbf{8{,}625.3} \pm 356.0$ | **+17%** |
| HalfCheetah (MuJoCo) | $\mathbf{17{,}523.0} \pm 33.2$ | $17{,}390.8 \pm 131.9$ | -1% |
| Hopper (MuJoCo) | $2{,}525.1 \pm 295.6$ | $\mathbf{3{,}693.1} \pm 85.6$ | **+46%** |
| HumanoidStandup (MuJoCo) | $160{,}203.0 \pm 1{,}357.2$ | $\mathbf{172{,}536.0} \pm 3{,}579.2$ | **+7%** |
| Swimmer (MuJoCo) | $127.7 \pm 2.7$ | $\mathbf{127.9} \pm 3.5$ | **+0.1%** |
| Walker2d (D4RL) | $\mathbf{88.0} \pm 1.7$ | $67.8 \pm 4.8$ | -23% |
| Ant (D4RL) | $145.4 \pm 0.2$ | $\mathbf{145.6} \pm 0.3$ | **+0.1%** |
| Halfcheetah (D4RL) | $58.3 \pm 0.1$ | $\mathbf{58.6} \pm 0.1$ | **+0.5%** |
| Hopper (D4RL) | $\mathbf{75.1} \pm 2.0$ | $71.2 \pm 3.2$ | -5.2% |
| | | | **+5.59%** |

Table 2: Comparison between TD7 and TD7+SQT.

| Environment | MaxMin | MaxMin+SQT | Improv. |
|---|---|---|---|
| Humanoid (MuJoCo) | $3{,}622.8 \pm 620.2$ | $\mathbf{3{,}680.6} \pm 631.1$ | **+1.6%** |
| Walker2d (MuJoCo) | $\mathbf{4{,}150.1} \pm 101.4$ | $3{,}781.8 \pm 278.7$ | -8.9% |
| Ant (MuJoCo) | $\mathbf{5{,}680.1} \pm 388.2$ | $5{,}364.7 \pm 412.0$ | -5.6% |
| HalfCheetah (MuJoCo) | $\mathbf{12{,}776.4} \pm 180.9$ | $11{,}030.1 \pm 707.7$ | -13.7% |
| Hopper (MuJoCo) | $3{,}608.0 \pm 4.2$ | $\mathbf{3{,}608.2} \pm 5.1$ | **+0.0%** |
| HumanoidStandup (MuJoCo) | $150{,}029.4 \pm 3{,}700.4$ | $\mathbf{152{,}634.3} \pm 3{,}318.2$ | **+1.7%** |
| Swimmer (MuJoCo) | $\mathbf{96.6} \pm 7.4$ | $94.5 \pm 8.2$ | -2.2% |
| Walker2d (D4RL) | $89.1 \pm 0.2$ | $\mathbf{90.8} \pm 0.2$ | **+1.9%** |
| Ant (D4RL) | $127.0 \pm 2.4$ | $\mathbf{130.6} \pm 1.5$ | **+2.8%** |
| Halfcheetah (D4RL) | $57.1 \pm 0.1$ | $\mathbf{58.1} \pm 0.3$ | **+1.7%** |
| Hopper (D4RL) | $87.7 \pm 1.2$ | $\mathbf{92.5} \pm 1.1$ | **+5.4%** |
| | | | **-1.39%** |

Table 3: Comparison between MaxMin and MaxMin+SQT.

| Environment | REDQ | REDQ+SQT | Improv. |
|---|---|---|---|
| Humanoid (MuJoCo) | $\mathbf{2{,}962.2} \pm 760.6$ | $2{,}196.1 \pm 640.5$ | -25.8 |
| Walker2d (MuJoCo) | $5{,}246.4 \pm 138.1$ | $\mathbf{5{,}545.6} \pm 147.3$ | **+5.7%** |
| HalfCheetah (MuJoCo) | $11{,}890.6 \pm 209.5$ | $\mathbf{12{,}230.0} \pm 133.9$ | **+2.9%** |
| Hopper (MuJoCo) | $1{,}639.5 \pm 150.2$ | $\mathbf{2{,}353.6} \pm 141.4$ | **+43%** |
| HumanoidStandup (MuJoCo) | $\mathbf{152{,}084.1} \pm 1{,}709.8$ | $151{,}845.8 \pm 3{,}605.7$ | -0.1% |
| Swimmer (MuJoCo) | $\mathbf{107.8} \pm 9.4$ | $91.3 \pm 10.1$ | -15.7 |
| Walker2d (D4RL) | $\mathbf{65.5} \pm 2.4$ | $54.4 \pm 4.4$ | -17% |
| Ant (D4RL) | $\mathbf{119.7} \pm 5.6$ | $110.3 \pm 4.7$ | -7.8 |
| Halfcheetah (D4RL) | $56.3 \pm 0.1$ | $\mathbf{57.2} \pm 0.1$ | **+1.6%** |
| Hopper (D4RL) | $\mathbf{76.8} \pm 2.2$ | $74.5 \pm 2.1$ | **-3%** |
| | | | **-1.62%** |

Table 4: Comparison between REDQ and REDQ+SQT.

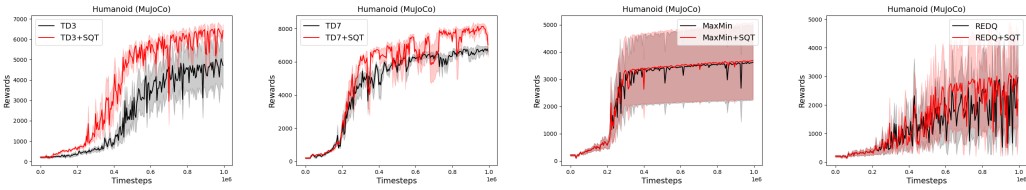

Figure 1: Results for Humanoid environment

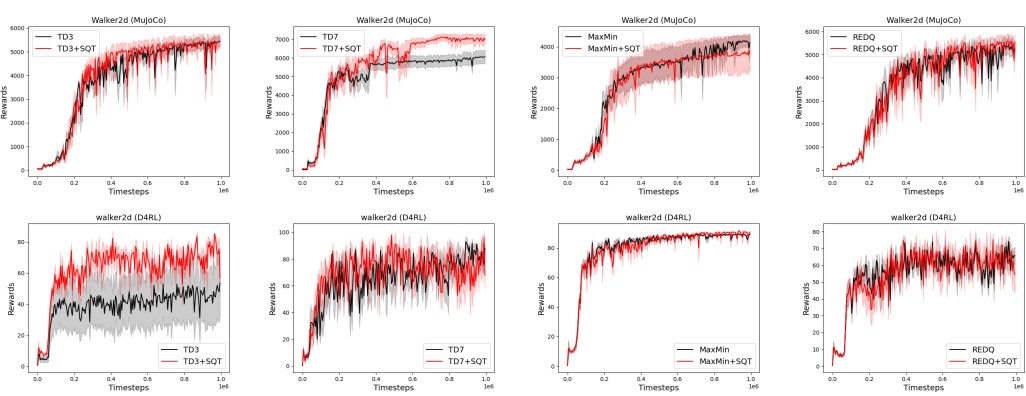

Figure 2: Results for Walker2D environment

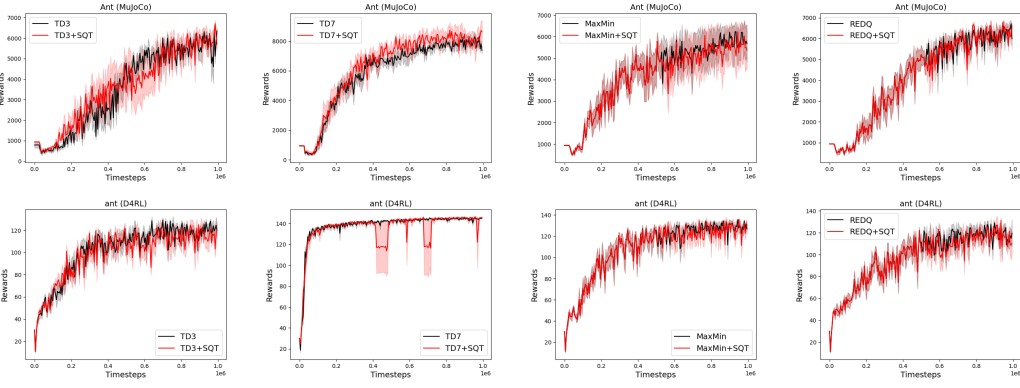

Figure 3: Results for Ant environment

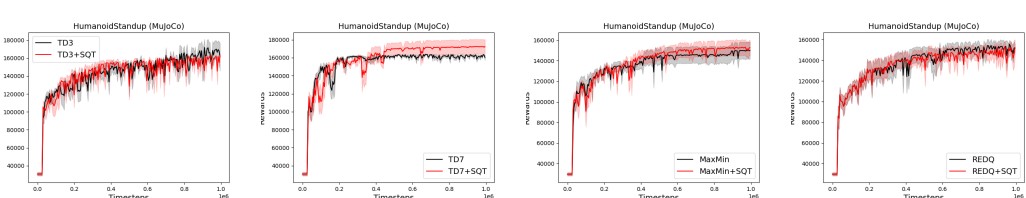

Figure 4: Results for Humanoid Standup environment

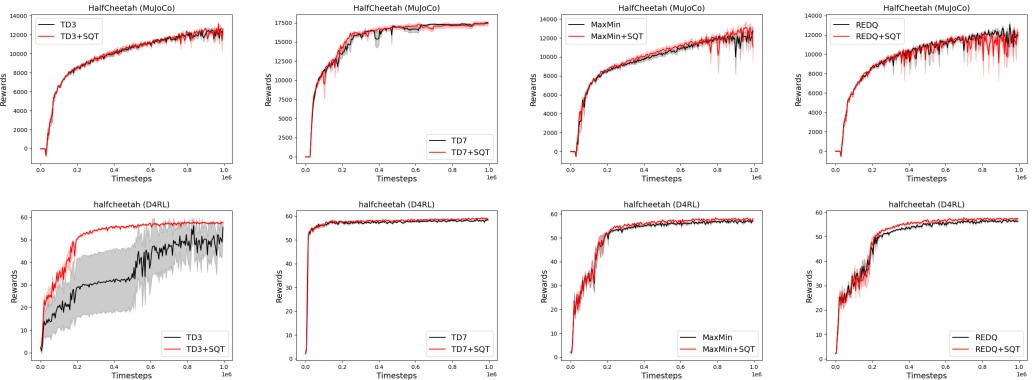

Figure 5: Results for HalfCheetah environment

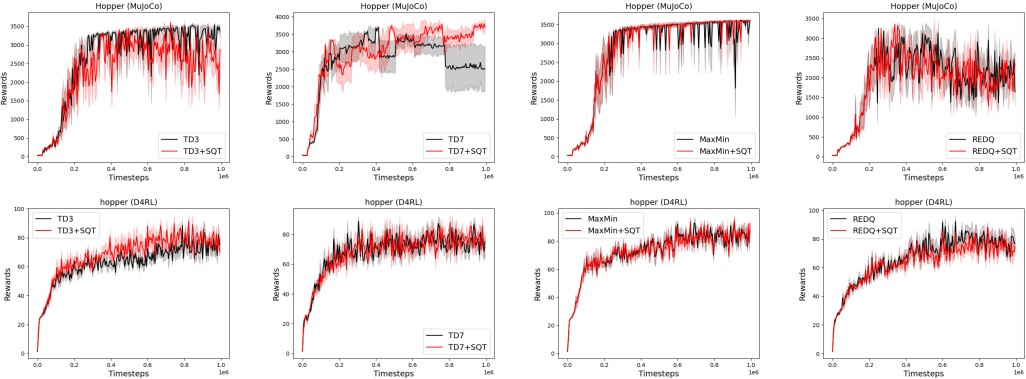

Figure 6: Results for Hopper environment

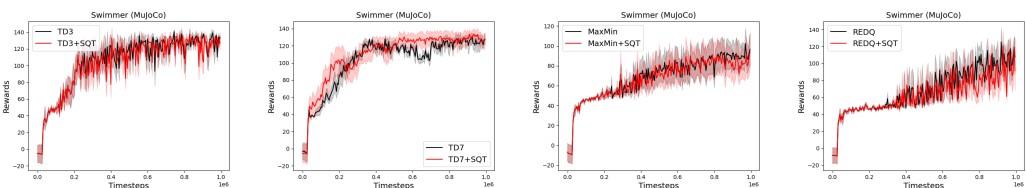

Figure 7: Results for Swimmer environment

## 6 DISCUSSION AND CONCLUSION

Std Q-target algorithm improves over SOTA continuous-space RL algorithms TD3, and TD7 on MuJoCo and D4RL Todorov et al. (2012); Fu et al. (2020) tasks on average. Our std Q-target formula has a clear advantage over TD3 or TD7 Q-target formula allowing the std Q-target algorithm to reach higher records on most of the tested tasks by reducing the unknown estimation area between the two Q-functions for safe values, pushing the algorithm to the real Q-values.

We note that our approach can be applied to any ensemble algorithm beyond the ones presented in this work, like REM (Agarwal et al., 2020), Bootstrap DQN (Osband et al., 2016), etc., as well as to discrete action algorithms. Also, in our algorithm there is a hyper-parameter $\alpha$ as well as a switching time where we add the regularization term. In future work, we propose to automate and schedule optimally these parameters.

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

# APPENDIX

## ALGORITHM IN PRACTICE

At the start learning steps of the algorithm, the Q-values are random, and thus, the Q-values Std can be large in comparison to the Q-network values and be considered as a random noise.

We tackle this problem by adding $t_{\text{switch}}$ hyper-parameter, reflecting initial total timesteps without applying the SQT formula, letting the Q-network stabilize at non-random values.

Practically, we compute the Q-target $y$ by the formula 11 if $t \geq t_{\text{switch}}$, and otherwise, by the original algorithm Q-target formula.

We use the following values in our experiments for $t_{\text{switch}}$ hyper-parameter:

| Algorithms | Environments | $t_{\text{switch}}$ |
|---|---|---|
| TD3, MaxMin, REDQ | Ant, Hopper | 500,000. |
| TD3, MaxMin, REDQ | Humanoid | 200,000. |
| TD3, TD7, MaxMin, REDQ | Humanoid Standup, Swimmer | 200,000. |

Table 5: Our experiments $t_{\text{switch}}$ hyper-parameter values.

In the rest of the experiments, we set $t_{\text{switch}}$ hyper-parameter as 100,000.

We use the following values in our experiments for $\alpha$ hyper-parameter:

| Algorithms | Environments | $\alpha$ |
|---|---|---|
| TD3, TD7, MaxMin, REDQ | Hopper | 0.01 |
| TD3, MaxMin, REDQ | Ant | 0.1 |

Table 6: Our experiments $\alpha$ hyper-parameter values.

In the rest of the experiments, we set $\alpha$ hyper-parameter as 1.

