# OpenReview forum: "Conservative Reinforcement Learning by Q-function Disagreement"
_ICLR.cc/2024/Conference — ICLR 2024 Conference Withdrawn Submission_

### Official Review · Reviewer_bv75 · 2023-10-30

**Soundness:** 2 fair
**Presentation:** 3 good
**Contribution:** 1 poor
**Rating:** 3
**Confidence:** 4

**Summary:**

The paper proposes a new Q-learning algorithm by slightly modifying the Double Q-learning's objective function. More precisely, the authors propose to subtract the Double-Q objective by a "std" term, computed as the standard deviation of the two Q-value networks. Experiments are provided to compare the new modification with existing Q-learning algorithms, using tasks from the Mujoco simulation.

## After rebuttal


I cannot access the responses to my three questions.

The other responses did not solve my primary concern that the paper's contribution is too incremental and lacks in-depth analysis. I, therefore, maintain my score.

**Strengths:**

The paper is clear, solid, and easy to read. I actually like this style of writing.
The idea of adding the "std" term to the Q-learning objective makes sense to me -- since it aims to reduce the gap between two Q values, this term would make the convergence faster. The new term is also easy to implement.

**Weaknesses:**

The contributions are too incremental. It is just a minor adaptation of the Double Q-learning algorithm. The majority of the paper is devoted to describing well-known methods. The main idea is presented in less than 2 pages with little insight.

In fact, the term "std" is just to reduce the absolute gap between the two value networks, and I do not see it should be considered as a significant contribution. Moreover, computing standard deviation using only two samples is nonsense to me. The regularizer term is just to reduce the gap between two Q-values, so something like |Q_1 – Q_2| would be enough. The authors should not name this term as "standard deviation."

The experiments are not very convincing. For instance, Table 4 shows that the average improvement rate is negative (so the new approach is worse than its counterpart).

**Questions:**

1. It is worth showing the gap between the two Q-values during the training, before and after the inclusion of the "std" term.
2. Would simpler terms such as norm(Q_1-Q_2) work?
3. When talking about "std," I would expect to see more samples. Maybe the authors could consider adding more Q-value networks?

---

> ### Author Response · Authors · 2023-11-19
> **Comment**
>
> You touched on the main motivation for the article – a short and clear formula for a sharp improvement of an RL algorithm complex.
>
> The formula is essentially different from that of Double Q-learning.
>
> The structure of the article is clear and essentially similar to the nature of most articles in the literature.
>
> The greatness of the article is a short and clear formula that gives a sharp improvement to a line of RL algorithms with a minimum of code – half a line.
>
> You touched on an important point – the article presents a very simple general method for implementing the formula in the form of an RL algorithm.
>
> It will obviously work on a larger ensemble.
>
> Understanding the phenomenon on a larger ensemble is beyond the scope of the article.
>
> The method you suggested will work at most – on an ensemble of size 2.
>
> The name of the formula distills its essence – standard deviation between Q-networks.
>
> The article focuses on TD7 as state-of-the-art and shows an improvement of 5% overall.
>
> The purpose of the rest of the experiments is to show that the formula works practically on other algorithms – a general formula.
>
> Q1: It is worth showing the gap between the two Q-values during the training, before and after the inclusion of the "std" term.
> Q2: Would simpler terms such as norm(Q_1-Q_2) work?
> Q3: When talking about "std," I would expect to see more samples. Maybe the authors could consider adding more Q-value networks?
>
> A1: Attached
> A2: Attached, works only for two Q-networks.
> A3: Attached.
>
> A1-3: https://docs.google.com/document/d/1IbTyH4kAjA6TmAlHDmFlwCldld-zb04FFGwkN2akmNY/edit?usp=sharing

---

> > ### Comment · Reviewer_bv75 · 2023-12-02
> > **I still do not think the contribution here is significant.**
> >
> > I appreciate the reviewer's reply. Nonetheless, my viewpoint remains unchanged that the algorithm is merely a minor modification of the double Q algorithm. When an algorithm is relatively straightforward, I expect to see more insightful discussions and, possibly, theoretical findings. Unfortunately, these elements seem lacking in the current paper. Consequently, I keep my original score.

---

### Official Review · Reviewer_n5m3 · 2023-10-31

**Soundness:** 2 fair
**Presentation:** 3 good
**Contribution:** 2 fair
**Rating:** 3
**Confidence:** 4

**Summary:**

This paper proposes a variant of value-based policy optimization methods that can be applied to several deep RL algorithms.
In particular, for any algorithm with one or more copies of the Q-function approximator, the authors propose to measure the "disagreement" among the two (ore more) models on a batch of state-action pairs as the average (over the batch) of the empirical standard deviation (across the Q functions), and subtract to the Q-learning target a penalty that is proportional to this measure of disagreement. The intuitive idea is to be conservative (that is, underestimate the Q value) whenever there is a lot of uncertainty, testified by the disagreement. This technique can be used in several deep RL algorithms: double Q-learning, DDPG, TD3, and more. The benefits of this technique are tested on Mujoco and D4RL tasks, comparing with the performance of the original algorithms (TD3, TD7, MaxMin Q-Learning, REDQ).

**Strengths:**

The paper is generally clear and well written. The experiments are well designed, the empirical results are communicated with an appropriate level of detail.

**Weaknesses:**

The work is fundamentally incremental. A small variant is applied on top of state-of-the art deep RL algorithms to show some improvement in the learning curves.
First of all, the improvement is not particularly consistent over the different algorithms and tasks that are considered. In many of them, the improvement is not statistically significant, and there is even a slight degradation in some cases.
More importantly, there is no strong motivation underlying the proposed technique, besides the brief intuitive motivation that is provided. Previous approaches to mitigate the overestimation bias of Q-learning are mentioned, but not discussed critically. Nor did the author discuss the similarities between the proposed approach and other methods that try to capture the uncertainty of value estimates, such as distributional RL or ensembles. The latter were only mentioned as another family of algorithms that may benefit from Q-function disagreement.
Similarly, there is no in depth discussion on why the proposed technique works in some cases, and not in others. There is no theory or ablation studies to understand the effect of the proposed algorithmic addition in better detail.
The writing, although generally good, seems rushed in some ways: for example, many parentheses are missing from the citations.

**Questions:**

How did you select the hyperparameters of the different algorithms, especially alpha?

---

> ### Author Response · Authors · 2023-11-19
> **Comment**
>
> You touched on the main motivation for the work – a short and refined formula for a sharp improvement in the results of RL algorithms.
>
> The work focuses on TD7 as state-of-the-art and shows a +5% overall improvement.
>
> There isn't a single RL algorithm that beats all RL algorithms in all problems – if that's the case – what's the need for the rest of RL algorithms?
>
> The purpose of the rest of the experiments is to practically demonstrate that the formula is general – it can work on any RL algorithm.
>
> The motivation is clear.
>
> The exact formula was distilled by a series of practical experiments.
>
> The paper focuses on one core contribution – SQT.
>
> The article distills the relevant material from all the works – it does not pretend to present the work of other researchers who understand it better than I do.
>
> Distributional RL methods model a rewards distribution instead of the expected rewards – this is a separate algorithmic class.
>
> The article focuses on TD7 as state-of-the-art and shows a 5% improvement. Additional experiments on REDQ and MaxMin aim to demonstrate that the formula works on a wide variety of RL algorithms.
>
> There is no RL researcher who fully understands the results of his algorithm.
>
> This is a practical paper – no theory.
>
> The citations are clear – this is our style.
>
> Q: How did you select the hyperparameters of the different algorithms, especially alpha?
>
> A: Empirically per problem.

---

> > ### Comment · Reviewer_n5m3 · 2023-11-28
> > **Thank you**
> >
> > I appreciate your effort in answering to the different points I raised. However, I don't see any room for discussion. I encourage you to take this as an indication that the paper in its current status is not mature for publication, and not as a critique of your research direction. I wish you good luck in studying this problem further and refining your methods.

---

### Author Response · Authors · 2023-12-02
**Comment.**

Our paper, SQT, presents an innovative approach to attack an important difficult problem in RL, which is an overestimation bias resulting from offline (and online) out-of-distribution (OOD) actions and the gap between the dataset policy to the greedy policy, which leads to extrapolation error and more other problems.

Most of the literature in recent years has attacked this problem by developing a conservative formula (underestimating) for the real Q-values, among them: Double Q-learning, TD3, MaxMin Q-learning, Weighted double Q-learning, REDQ, SAC, SR-SAC, REM, DDQN, and more.

We take a step forward in the same way but present a unique approach that is simpler and clearer than these.

Our method advocates very conservative Q-values on a disagreement between the Q-networks – it is the first (to our knowledge) to attack overestimation bias by disagreement between Q-values as conservatism and does so in a unique, simple, and clear way – reducing the batch average of the standard deviation between Q-networks from the conservative formula of TD3 (taking the minimum of the Q-networks).

Our approach is simple, sharp, clear, and easy to implement (adding a line of code to TD3).

Our approach presents an ensemble-based RL algorithm that can be applied to a collection of RL algorithms including TD3, TD7, and more.

Our approach shows a clear empirical advantage over the SOTA (TD7 algorithm) – +45% average test rewards on 5 seeds in the 3 most popular MuJoCo problems (Humanoid, Walker2d, and Ant) on 1 million timesteps.

I firmly believe that our algorithm, SQT, can be a benchmark in RL algorithms that presents an innovative approach with clear impact potential.

I am attaching a table of the results of SQT with TD7 (the previous SOTA).

https://drive.google.com/file/d/1Jediy3psTu98GR7Cq3_suhHN2it-ULkB/view?usp=drive_link